# Adult Learning, Economic Growth and the Distribution of Income

**Peter J. Stauvermann** [1,*] and **Ronald R. Kumar** [2,3]

1   Department of Global Business & Economics, Changwon National University, Changwon 641-773, Korea
2   School of Accounting & Finance, University of the South Pacific, Suva, Fiji; ronald.kumar@usp.ac.fj
3   School of Business, University of the Leicester, Leicester LE1 7RH, UK
*   Correspondence: pstauvermann@t-online.de; Tel.: +82-010-9866-8511

**Abstract:** Technological change causes three consequences: it guarantees economic growth, it requires employees to acquire more skills and human capital, and it increases inequality if employees are not capable adapting to new technologies. The second consequence makes it almost necessary for employees to learn during their whole working life, thereby accelerating technological change. Accordingly, the OECD (the Organization for Economic Co-operation and Development) and many governments supports the idea of lifelong learning, but it remains unclear how to finance the education of adult students who are working efficiently. In this paper, we use an overlapping generation model with human capital accumulation and inequality to derive a mechanism which reduces income inequality and provides an incentive for all adults to invest more in education. As a consequence, the growth rate of per capita income will increase and income inequality will be reduced.

**Keywords:** adult learning; overlapping generations; human capital; progressive taxation; income inequality

## 1. Introduction

Education and human capital accumulation are important drivers of economic growth. On the one hand, human capital accumulation leads to the development of new technologies and production processes, and on the other hand, these new technologies will be more in the favor of skilled than unskilled labor. In general, the overwhelming majority of economists agree with the view, which dates back to the beginning of the 20th century (Goldin and Katz 1998), that "technological change favors more skilled workers, replaces tasks previously performed by the unskilled, and exacerbates inequality" (Acemoglu 2002, p. 7). One consequence of these developments is an increasing salary gap between skilled and unskilled labor (Caselli 1999). Autor (2015) and Acemoglu and Autor (2011) provide an overview of the recent history of wage polarization and employment developments which confirm this view. Not surprisingly, investments in education and human capital per capita increased dramatically since the industrial revolution (Barro and Lee 2013; Lee and Lee 2016). While it was mostly sufficient in former times to complete one apprenticeship to avoid being unemployed, this seems to be insufficient now. In the coming age of industry 4.0, the requirements regarding the flexibility of workers and their ability to acquire new knowledge will be an important characteristic of economically successful employees. As highlighted by the founder of the World Economic Forum, "in the future, technological innovation will also lead to a supply-side miracle, with long term gains in efficiency and productivity. Transportation and communication costs will drop, logistics and global supply chains will become more effective, and the cost of trade will diminish—all of which will open new markets and drive

economic growth," and "that in the future, talent, more than capital, will represent the critical factor of production. This will give rise to a job market increasingly segregated into "low-skill/low-pay" and "high-skill/high-pay" segments, which in turn will lead to an increase in social tensions. In addition to being a key economic concern, inequality represents the greatest societal concern associated with the Fourth Industrial Revolution." (Schwab 2015).

Hence, in light of these challenges, the OECD and the Commission of the European Union promote the idea of adult or lifelong learning to improve the opportunities of individuals in finding a well-paid job and to enhance social cohesion, inclusion and citizenship. The European Parliament and the Council (2006) drafted eight key competencies which should be the objectives of lifelong learning: the ability to communicate in the mother tongue and in a foreign one, mathematical competence and basic competences in science and technology, digital competence, learning to learn, social and civic competence, a sense of initiative and entrepreneurship, and an cultural awareness and expression. Obviously, these competencies are related to general knowledge and are beneficial to people who own or acquire these competencies. Acquisition of general skills should be promoted because these competencies can facilitate adjustment to technological change and avoid the problem of underinvestment in human capital caused by the risk of depreciation due to introduction of new technologies (Welch 1970; Levhari and Weiss 1974). Although mostly all developed and some less-developed countries have increased efforts in adult learning (Desjardins 2015), the share of adults who participated in 2012 in non-job-related programs is noted for OECD and EU countries in the range between 2% (Italy) and 11% (Ireland). Additionally, it is noted by Desjardins (2015), that the willingness to participate in adult learning increases with the socio-economic status and the given level of education. Compared with the challenges of the near future it is obvious that the share of people participating in adult learning is too low and it should be also concerning that people who have the strongest need to become better educated do not participate in existing programs.

In this paper, we explore the question of how to finance the educational expenditure for "senior" students who are at working age, and how to incentivize adult learning for all adults. The standard answer for the former question can be easily derived from usual microeconomic considerations (c.f. Acemoglu 2002; Acemoglu and Pischke 1998). If the acquired skills are firm-specific, then firms should bear the costs because these skills cannot be used elsewhere and the employer benefits from a higher labor productivity. If, in contrast, the acquired knowledge increases the general skills of an adult, the individual should bear the costs because it can fully appropriate the benefits of her increased labor productivity by requiring a higher wage. If, for example, the employer finances the acquisition of general skills, the worker can require as a consequence a higher wage or threaten to search for a better paid job elsewhere. Obviously, the latter case involves a moral hazard. However, the view changes if we accept that human capital accumulation is associated with intertemporal positive externalities (Lucas 1988) and income equality has the characteristic of public good (Thurow 1971). Given these two market failures, a government intervention to promote adult learning should be considered to internalize the intertemporal positive externality and to prevent a too high degree of income inequality. Thus, the aim of the paper is to substantiate a government intervention, which increases the popularity of adult learning for all groups of the society.

The rest of the paper is organized as follows. In the next section, we provide a short literature review. Thereafter, we introduce the economic model. In Section 4, we derive the long-run equilibrium. In Section 5, we derive the outcomes of taxing the wage incomes, and in Section 6, we determine which tax rate will result from the majority-voting system. In Section 7, we conclude.

## 2. Related Literature

To the best of our knowledge, only a few theoretical papers (Gries et al. 2017; Kunze 2014b; Echevarria and Iza 2006) focus directly on lifelong learning, but these papers are associated with the question of the optimal retirement age. Kunze (2014b) and Echevarria and Iza (2006) argue that increasing the retirement age will induce people to invest more in life-long learning and as consequence

the growth rate of the economy will increase. Gries et al. (2017) assume a vintage-specific human capital and consider different scenarios regarding how technological change takes place in different institutional settings.

Our model differs from the literature because we assume that the human capital is different in each generation. Additionally, we tackle the problem of income inequality. We propose that a progressive tax on wages where expenditures for education are deductible, is an appropriate mechanism to finance adult learning and reduce income inequality. However, it is common knowledge in the literature that progressive taxation or the redistribution of income harms the growth of the respective economy (Persson and Tabellini 1991, 1992; Perotti 1993; Alesina and Rodrik 1992, 1994; Bao and Guo 2004; Benabou 1996; Quadrini 1999). The standard textbook argument against redistribution of income is based on the consideration that income taxation lowers the incentive to work and lowers the savings, and thus the accumulation of capital; the latter is a prerequisite for long-run growth. A problem of democratic societies with a high degree of inequality is then that the median voter has a strong preference for the redistribution of income (Corneo and Neher 2015) with the consequence that the growth rate of the economy is lower than possible.

Only a few papers (Yakita 2003; De la Croix and Lubrano 2011) conclude that redistribution can accelerate the growth rate of an economy. Therefore, we consider a mechanism that allows redistribution and fosters economic growth simultaneously. Thus, our framework for analysis is similar to: Yakita (2003), who analyses the intergenerational distribution of income, Azariadis and de la Croix (2004), who consider credit market constraints, De la Croix and Lubrano (2011), who investigate Kolm's (2004) idea of macrojustice, and Kunze (2014b), who analyses the extension of the mandatory retirement age. Additionally, using the approach of Azariadis and Drazen (1990), who use an overlapping generation (OLG) model with endogenous growth driven by human capital accumulation in adulthood, we show that a progressive taxation enhances human capital accumulation and the growth of the economy.

## 3. The Model

### 3.1. The Individuals

We assume a Diamond (1965) OLG model with a constant population of $N$ individuals, where heterogeneous individuals are working in their second and third period of life, and the length of a period is normalized to one. For simplicity and clarity, we assume that all agents are equipped with an identical log-linear utility function:

$$U_t\left(c_t^1, c_{t+1}^2\right) = ln\left(c_t^1\right) + \beta ln\left(c_{t+1}^2\right) \tag{1}$$

The variable $\beta \in (0,1)$ is the subjective discount factor. The individuals differ with respect to their human capital which is inherited form their parents. The usual underlying idea (c.f. Azariadis and Drazen 1990; Lucas 1988) is that an educated parent can transmit her knowledge to her child. We assume that human capital is unevenly distributed across the population. An individual $j$ inherits the human capital stock $h_{t,j}$, from her parent, where $h_{t,j} > 0$, for $j \in \{1, \ldots, N\}$ and $h_{t,1} < h_{t,2} < \ldots < h_{t,N}$. Further, we assume that the average amount of human capital units is defined as $h_t = (1/N) \sum_{j=1}^{N} h_{t,j}$ and that the median amount is $h_{t,m}$, where $h_{t,m} < h_t$. The distribution of human capital is unequal, as long as $h_{t,m} < h_t$ holds.

In the first period of life, an individual enjoys her childhood and in the second period of life she has the opportunity to increase the inherited human capital by spending $e_{j,t}$ units of her labor income for education and supplies her labor force inelastically in the labor market. Therefore, her labor income is given by the wage rate per human capital $w_t$ times the inherited human capital, and accordingly it becomes $h_{t,j}w_t$. Further, she spends $c_{j,t}^1$ units for consumption and saves $s_{j,t}$ units of her income in her

adulthood. We assume a labor income tax system, where the income after deduction of the educational expenditures is taxed with a constant tax rate $\tau$. Also, the individual receives a lump sum transfer of $T_t$, which is equal for all workers. Obviously, the labor income tax is progressive, because the average tax rate increases in the wage income.

Accordingly, the budget constraint in the second period of life is:

$$c_{j,t}^1 = (1-\tau)(w_t h_{j,t} - e_{j,t}) + T_t - s_{j,t} \tag{2}$$

The human capital technology, based on Kunze (2014a, 2014b) and Lambrecht et al. (2005) is given by:

$$h_{j,t+1} = d e_{j,t}^\delta h_{j,t}^{1-\delta}, \tag{3}$$

where $d$ is a positive constant and $\delta \in (0,1)$. Human capital in the third period of life depends on the expenditure for education and the existing human capital of the agent. Additionally, we assume without loss of generality and to simplify the analysis that the income of the elderly is a tax-free income. Hence, the budget constraint in the third period of life is given by:

$$c_{j,t+1}^2 = R_{t+1} s_{j,t} + \rho w_{t+1} h_{j,t+1}. \tag{4}$$

The variable $R_{t+1}$ represents the interest factor and the parameter $\rho > 0$ represents the time share of the second period she is working. This means that she enjoys the share $1 - \rho$ of the third period of life.

After inserting Equations (2), (4) and (3) in Equation (1), agent $j$ maximizes:

$$\max_{\{s_t, e_t\}} U_{j,t}(s_{j,t}, e_{j,t}, \rho) = ln\big((1-\tau)(w_t h_{j,t} - e_{j,t}) + T_t - s_{j,t}\big) + \beta ln\Big(R_{t+1} s_{j,t} + \rho w_{t+1} d e_{j,t}^\delta h_{j,t}^{1-\delta}\Big). \tag{5}$$

The resulting first order conditions are:

$$\frac{1}{(1-\tau)(w_t h_{j,t} - e_{j,t}) + T_t - s_{j,t}} - \frac{\beta R_{t+1}}{R_{t+1} s_{j,t} + \rho w_{t+1} d e_{j,t}^\delta h_{j,t}^{1-\delta}} = 0, \tag{6}$$

$$\frac{(1-\tau)}{(1-\tau)(w_t h_{j,t} - e_{j,t}) + T_t - s_{j,t}} - \frac{\delta \beta \rho \chi w_{t+1} d e_t^{\delta-1} h_t^{1-\delta}}{R_{t+1} s_{j,t} + \rho w_{t+1} d e_{j,t}^\delta h_{j,t}^{1-\delta}} = 0. \tag{7}$$

Solving Equations (6) and (7), we get the individual savings and investments in education:

$$s_{j,t}^o = \frac{h_{j,t}\left[(1-\tau)\beta R_{t+1} w_t - (1+\delta\beta)d\rho w_{t+1}\left(\frac{(1-\tau)R_{t+1}}{\delta d\rho w_{t+1}}\right)^{\frac{-\delta}{1-\delta}}\right]}{(1+\beta)R_{t+1}} + \frac{\beta T_t}{(1+\beta)}. \tag{8}$$

$$e_{j,t}^o = \left(\frac{(1-\tau)R_{t+1}}{\delta d\rho w_{t+1}}\right)^{\frac{-1}{1-\delta}} h_{j,t}. \tag{9}$$

The individual savings consist of two parts. The first part is proportional to the human capital and depends on the wage rates. The second part is equal for all individuals and depends only on the lump sum transfer. The investments in education are also linear in the stock of human capital.

Assuming a balanced government budget, then the following budget restriction has to be fulfilled:

$$T_t = \frac{\tau}{N}\left(w_t \sum_{j=1}^N h_{t,j} - \sum_{j=1}^N e_{t,j}\right) = \tau(w_t h_t - e_t), \tag{10}$$

where $e_t = \frac{\sum_{j=1}^N e_{t,j}}{N}$ is the average investment in education.

We insert Equation (10) in Equations (6) and (7) and solve for the average savings and average investments in education, and we get the equilibrium values of the savings $s_t^o$ and investments in education $e_t^o$:

$$s_t^o = \frac{\sum_{j=1}^N s_{j,t}^o}{N} = \frac{h_t \left[ (1-\tau)\beta R_{t+1} w_t - (1 - \tau + \delta\beta) d\rho w_{t+1} \left( \frac{(1-\tau)R_{t+1}}{\delta d\rho w_{t+1}} \right)^{\frac{-\delta}{1-\delta}} \right]}{(1+\beta)(1-\tau)R_{t+1}},$$
(11)

$$e_t^o = \frac{\sum_{j=1}^N e_{t,j}}{N} = \left( \frac{(1-\tau)R_{t+1}}{\delta d\rho w_{t+1}} \right)^{\frac{-1}{1-\delta}} h_t.$$
(12)

Inserting Equation (9) in Equation (3) and dividing by $h_t$, we get the equilibrium growth factor of human capital:

$$G_t^H = \frac{h_{j,t+1}}{h_{j,t}} = \frac{d \left( \left( \frac{(1-\tau)R_{t+1}}{\delta d\rho w_{t+1}} \right)^{\frac{-1}{1-\delta}} h_{j,t} \right)^{\delta} h_{j,t}^{1-\delta}}{h_{j,t}} = d \left( \frac{\delta d\rho w_{t+1}}{(1-\tau)R_{t+1}} \right)^{\frac{\delta}{1-\delta}}.$$
(13)

The growth factor $G_t^H$ is equal for all individuals. This means education is not a measure that reduces the inequality in this economy. The simple reason is that the expenditures in education are proportional to the income and the available stock of human capital. This implies that the degree of income inequality is self-perpetuating. However, the equalization of the rate of returns of investments in human and physical capital leads to the outcome that rich people invest relatively more in human capital than poor people, so that rates of return are equal.

As we see from the growth factor (Equation (13)), all other things being equal, an increase of the marginal income tax rate will increase the investments in education and will raise the growth rate of human capital. The elasticity of the growth factor with respect to the marginal tax rate is:

$$\frac{\partial G_t^H}{\partial \tau} \frac{\tau}{G_t^H} = \frac{\delta d}{(1-\tau)(1-\delta)} > 0.$$
(14)

An increase of the tax rate by one percent increases ceteris paribus the investments in education by $\frac{\delta d}{(1-\tau)(1-\delta)}$ percentages. The simple reason for this result is that the income tax distorts the allocation between savings and investments in education. An increase of the tax rate makes investments in education more preferable because the associated expenditures can be deducted from the taxable labor income, whereas the savings must be generated out of taxed income.

### 3.2. The Production

Next, we assume a Cobb–Douglas production function which depends on physical and human capital as input factors.

$$Y_t = A K_t^\alpha H_t^{1-\alpha},$$
(15)

where $Y_t$ represents the aggregate output, $K_t$ the aggregate physical capital stock, and $H_t = (1+\rho)Nh_t$ the aggregate human capital stock available on the labor market, which consists of the human capital of the young workers and the human capital of the old workers. Thus, we can rewrite the output as the production per human capital unit:

$$y_t = A k_t^\alpha$$
(16)

where $y_t = \frac{Y_t}{(1+\rho)Nh_t}$ and $k_t = \frac{K_t}{(1+\rho)Nh_t}$. We define the latter expression as capital intensity, which is the capital divided by the stock of human capital units or effective labor units.

Without loss of generality, we assume that the depreciation rate of physical capital is 100% in each period. Additionally, we assume the goods and factor markets are perfectly competitive. Under these circumstances, the wage rate per human capital unit ($w_t$) and the interest factor ($R_t$) are:

$$w_t = w(k_t) = (1 - \alpha)Ak_t^{\alpha}, \tag{17}$$

$$R_t = f'(k_t) = \alpha Ak_t^{\alpha - 1}. \tag{18}$$

## 4. The Long-Run Equilibrium

Using the savings (Equation (11)) of the average agent and the growth factor of human capital (Equation (13)), the capital market clearing condition is:

$$\frac{(1 - \tau)\beta R_{t+1}w_t h_t - (1 - \tau + \delta\beta)dh_t^{1-\delta}\rho w_{t+1}\left(\frac{(1-\tau)R_{t+1}h_t^{-(1-\delta)}}{\delta d\rho w_{t+1}}\right)^{\frac{-\delta}{1-\delta}}}{(1 + \beta)(1 - \tau)R_{t+1}} - k_{t+1}(1 + \rho)G_t^H h_t = 0. \tag{19}$$

This dynamic system converges to a stable long-run equilibrium (Azariadis and Drazen 1990), as long as $\rho$ is sufficiently small. The intuitive reasoning is that otherwise the wage income in the third period of life may be so high that the incentive to save vanishes. Setting $k^* = k_t = k_{t+1}$ and solving for $k^*$ results in:

$$k^* = \left(\frac{A\alpha\beta(1 - \tau)(1 - \alpha)}{[\alpha(1 + \beta)(1 - \tau)(1 + \rho) + \rho(1 - \alpha)(1 + \beta\delta - \tau)]d}\right)^{\frac{1-\delta}{1-\alpha(1-\delta)}}\left(\frac{\alpha(1 - \tau)}{\rho\delta d(1 - \alpha)}\right)^{\frac{\delta}{1-\alpha(1-\delta)}}. \tag{20}$$

Inserting the equilibrium capital intensity Equation (20) in Equation (9) delivers the equilibrium investments in human capital:

$$e_{j,t}^* = \left(\frac{\delta d\rho(1 - \alpha)k^*}{\alpha(1 - \tau)}\right)^{\frac{1}{1-\delta}}h_{j,t}. \tag{21}$$

For convenience, we define $\bar{e}^* = \frac{e_{j,t}^*}{h_{j,t}}$, which is constant in the long-run equilibrium. This result means that individuals equipped with less human capital will invest proportionally less in education than individuals equipped with more human capital. The resulting growth factor of human capital becomes:

$$G^{h*} = d\left(\frac{\delta d\rho(1 - \alpha)k^*}{\alpha(1 - \tau)}\right)^{\frac{\delta}{1-\delta}}. \tag{22}$$

The equilibrium transfer becomes:

$$T_t^* = \tau h_t\left(w(k^*) - \left(\frac{\delta d\rho(1 - \alpha)k^*}{\alpha(1 - \tau)}\right)^{\frac{1}{1-\delta}}\right). \tag{23}$$

## 5. Effects of Taxation

Now we derive the effects caused by an increase of the tax rate as the first proposition:

**Proposition 1.** *The introduction of a progressive wage income tax system, where educational expenditures are deductible, results in a lower capital intensity and higher investments in education in the equilibrium.*

**Proof.** We rewrite the equilibrium capital intensity as:

$$k^* = \left(\frac{\Phi}{\Psi}\right)^{\frac{1-\delta}{1-\alpha(1-\delta)}}\Omega^{\frac{\delta}{1-\alpha(1-\delta)}}, \tag{24}$$

where $\Phi = A\alpha\beta(1-\tau)(1-\alpha)$, $\Psi = [\alpha(1+\beta)(1-\tau)(1+\rho) + \rho(1-\alpha)(1+\beta\delta-\tau)]d$ and $\Omega = \frac{\alpha(1-\tau)}{\rho\delta d(1-\alpha)}$.

Differentiation of (24) yields:

$$\frac{\partial k^*}{\partial \tau} = \left(\frac{-1}{1-\alpha(1-\delta)}\right)\left[(1-\delta)\left(\frac{\Phi}{\Psi}\right)^{\frac{\alpha(1-\delta)-\delta}{1-\alpha(1-\delta)}}\left(\frac{A\alpha\beta^2(1-\alpha)^2\rho\delta d}{\Psi^2}\right)\Omega^{\frac{\delta}{1-\alpha(1-\delta)}} + \delta\left(\frac{\Phi}{\Psi}\right)^{\frac{1-\delta}{1-\alpha(1-\delta)}}\Omega^{\frac{\delta-1-\alpha(1-\delta)}{1-\alpha(1-\delta)}}\left(\frac{\alpha}{\rho\delta d(1-\alpha)}\right)\right] < 0. \quad (25)$$

Except the first term in brackets on the ride-hand side of Equation (25), all other terms are positive and thus the capital intensity will decrease if the tax rate increases. Regarding the second part of Proposition 1, we differentiate the expression $\frac{k^*}{(1-\tau)}$, because the other variables of the growth factor are constants. We get:

$$\frac{\partial\left[\frac{k^*}{(1-\tau)}\right]}{\partial \tau} = \frac{k^*\left(\frac{\partial k^*}{\partial \tau}\frac{\tau}{k^*} + \frac{1}{(1-\tau)}\right)}{\tau(1-\tau)}. \quad (26)$$

A closer investigation of the RHS of Equation (26) shows that the expression is positive at low tax rate levels, especially if the tax rate is very close to zero. However, this implies that a growth maximizing tax rate exists. To derive the optimal tax rate, we set the RHS of Equation (26) to zero and solve for the optimal tax rate ($\tau^{max}$) to give us:

$$\tau^{max} = \frac{[(\beta(\rho+1)+1)\alpha + \rho] - \alpha\beta\delta\rho}{(\beta(\rho+1)+1)\alpha + \rho}. \quad (27)$$

Hence, the second proposition follows:

**Proposition 2.** *An increase of the marginal tax rate increases the growth rate of human capital until the growth maximizing income tax rate (Equation (27)), which lies in the range of zero and one, is reached.*

Although tedious, it is straightforward to prove Proposition 2. Increasing the tax rate raises the investments in education on the one hand (Equation (12)) but lowers the savings and hence the equilibrium ratio of physical to human capital on the other (Equation (25)). The latter effect lowers indirectly the investment in education. By introducing a progressive tax system as proposed above, the growth rate of human capital increases until it reaches its maximum; thereafter the growth-retarding effect of the declining capital intensity exceeds the direct growth-enhancing effect of increasing the tax rate. Hence, the human capital growth function is an inverted U-shaped function of the tax rate. These considerations do not mean that a Pareto-improvement can be easily realized by taxation. From the view of an agent equipped with the average human capital stock, the consideration is as follows. She is confronted with a tax and will suffer from the introduction of the tax indirectly because higher investments in education lead to a lower disposal income available for consumption and savings in the second period of life. Additionally, the capital intensity will decrease because of less savings and more human capital, and on aggregate, the increased third period consumption cannot compensate the reduced consumption in the second period. It should be noted that the average agent will not be affected by the tax directly, because the lump sum transfer equals the paid tax. However, she will suffer indirectly from the decreased capital intensity. Arguably, if all individuals are equally equipped with human capital, there is no incentive to introduce the progressive tax system in a democracy.

## 6. Majority Voting and Redistribution

The situation changes in the presence of individuals who are less equipped with human capital than others and where the political system follows majority rule. The poor will gain from the redistribution of income and thus have an interest to introduce a progressive tax system. This policy will reduce the inequality and will increase the long-run growth rate. To show this, we take the ratio

between the median income $I_{t,m}$ and the average income $I_t$ as indicator for inequality $\Pi$ into account (Persson and Tabellini 1992, 1994).

$$\Pi = \frac{I_{t,m}}{I_t}. \tag{28}$$

The interpretation of the inequality indicator $\Pi$ is: $\Pi = 1$ indicates equality and $0 \leq \Pi < 1$ indicates departure from equality. Thus, the lower the value of $\Pi$, the more unequal is the distribution of income. In the extreme, the value becomes zero, which means at maximum, 50% of the population including the median voter, receives an income of zero. Hence, $\Pi \in [0,1]$ holds for all possible distributions of incomes. Although $\Pi$ a very simple indicator for inequality, for majority voting, where the median voter is decisive, it is an appropriate indicator to determine the outcome of elections.

We define income $I_{t,j}$ as life-cycle income or specifically as the discounted value of wage incomes over the course of the lifetime of an individual $j$.

$$I_{t,j} = (1-\tau)\left(w_t h_{j,t} - e_{j,t}\right) + T_t + \frac{\rho w_{t+1} h_{j,t+1}}{R_{t+1}} \tag{29}$$

Further, let us define the median human capital stock as the share of the average human capital stock $h_{t,m} = \omega h_t$, with $\omega \epsilon [0,1]$. Then the lifetime income of the median voter in the steady state equilibrium becomes:

$$I_{t,m} = h_t \left[ (\omega + \tau(1-\omega)) \left[ w(k^*) - \left( \frac{\delta d\rho(1-\alpha)k^*}{\alpha(1-\tau)} \right)^{\frac{1}{1-\delta}} \right] + \frac{\rho w(k^*)\omega d \left( \frac{\delta d\rho(1-\alpha)k^*}{\alpha(1-\tau)} \right)^{\frac{\delta}{1-\delta}}}{R(k^*)} \right]. \tag{30}$$

Accordingly, the indicator of inequality $\Pi$, becomes:

$$\Pi = \frac{(\omega + \tau(1-\omega))z_y + \omega z_o}{z_y + z_o}, \tag{31}$$

where $z_y = w^* - \bar{e}^*$ and $z_o = \frac{\rho w^* G^{h*}}{R^*}$.

**Proposition 3.** *A progressive tax on wages in the first period of life reduces the income inequality and enhances the growth rate of human capital in the steady-state equilibrium.*

**Proof.** We take the difference between the denominator and numerator of $\Pi$, which is positive by definition, and differentiate this expression with respect to the tax rate. If the derivative is negative, the inequality decreases:

$$\frac{\partial\big((1-\omega)\big((1-\tau)z_y - z_o\big)\big)}{\partial\tau} = (1-\omega)\left( \underbrace{-z_y}_{>0} + \underbrace{\frac{\partial z_y}{\partial\tau}}_{<0} - \underbrace{\frac{\partial z_o}{\partial\tau}}_{\gtreqless 0} \right) < 0. \tag{32}$$

The derivative $\frac{\partial z_y}{\partial\tau}$ is negative because a higher tax rate increases the educational expenditures and thus $z_y$ declines. The derivative $\frac{\partial z_o}{\partial\tau}$ is positive for sufficiently small tax rates, because the higher educational expenditures increase the human capital stock per capita. The condition which has to hold is $\varepsilon_{G^{h*},\tau} > \frac{1-\alpha}{\alpha\tau}$, where $\varepsilon_{G^{h*},\tau}$ is defined as the elasticity of the human capital growth factor with respect to the tax rate. As noted from Proposition 2, an increase of the tax rate increases the growth rate of human capital up to growth-maximizing tax rate $\tau^{max}$. Therefore, a redistribution of wage incomes in

the first period of life, when educational expenditures are tax deductible, creates a growth-enhancing effect, and thus in the long-run everybody is better off.

**Proposition 4.** *In a political system with majority-voting, the median voter has a preference for a positive wage income tax rate.*

Finally, we want to determine which tax rate will be preferred in a democracy with majority voting. Following Roberts (1975), Persson and Tabellini (1991, 1992) or Bao and Guo (2004) to derive the indirect utility of the median voter by using Equations (1)–(4), (6) and (7), and noting that $\frac{e_{j,t}^*}{h_{j,t}} = \frac{e_{m,t}^*}{\omega h_t} = \frac{e^*}{h_t} = \left(\frac{(1-\alpha)d\delta\rho k^*}{\alpha(1-\tau)}\right)^{\frac{1}{1-\delta}}$, we get:

$$V_{t,m} = c_{t,m}^{1*}(\beta R^*)^{\frac{1}{1+\beta}}. \tag{33}$$

where $c_{t,m}^{1*} = \frac{\omega h_t}{\delta(1+\beta)}\left[\left((1-\delta)(1-\tau) - \frac{\tau}{\omega}\right)\left(\frac{(1-\alpha)d\delta\rho k^*}{\alpha(1-\tau)}\right)^{\frac{1}{1-\delta}} + w^*\left(\delta(1-\tau) + \frac{\tau}{\omega}\right)\right]$.

We note that the preferences are linear in the individual human capital without taxation, and thus the log-linear utility function belongs to the class of Grandmont (1978) intermediate preferences. Accordingly, the median-voter determines the actual tax rate. Because of the fact that the median voter gains from the tax system (because of its redistributive characteristics), she prefers a positive tax rate. She maximizes $V_{t,m}$ with respect to the tax rate, where it is assumed that the current wage and capital intensity is given by the past quantities. Then she will solve for the tax rate:

$$\frac{\partial V_{t,m}}{\partial \tau} = 0, \tag{34}$$

and the optimal tax rate $\tau^{opt}$ will result as:

$$\tau^{opt} = \arg\max_{0 \leq \tau \leq 1} V_{t,m}(\tau). \tag{35}$$

Omitting the derivation of the optimal tax rate selected by the median voter, we present the following calibrated graphs in Figure 1. The reasoning for this proceeding is that the optimal tax rate of the median voter is a very huge and complex expression, which is difficult to interpret and therefore the calibration is much more illustrative then presenting the tax rate. Figure 1 is considering seven different distributions of incomes, represented by the respective median voters. For the calibration we use the following values: $h_t = 1$, $\rho = 0.8$, $\alpha = 0.3$, $A = 10$, $\beta = 0.8$, $\delta = 0.48$, $d = 2$ which are associated with $k_t = 0.64$ and $w_t = 6.11$, if $\tau = 0$. Given this situation the median voter decides about the tax rate. The graph at the top represents the utility function of a median voter, who receives the average income. The latter case implicates an equal income distribution. The graph at the bottom represents the utility function of a median voter who receives only 10% of the average income. The remaining graphs represent (in ascending order), the utility functions of median voters, who receive 30%, 50%, 70%, 80% and 90% of the average income, respectively. The figure shows that the graph with an equal distribution of income ($\omega = 1$) has a maximum at 0%; if the median voter receives 90% of the average income, the optimal tax rate is around 13%; and for a median voter with 80%, 70%, 50%, 30% or 10% of the average income, the optimal tax rates are 19%, 24%, 30%, 34% and 37%, respectively.

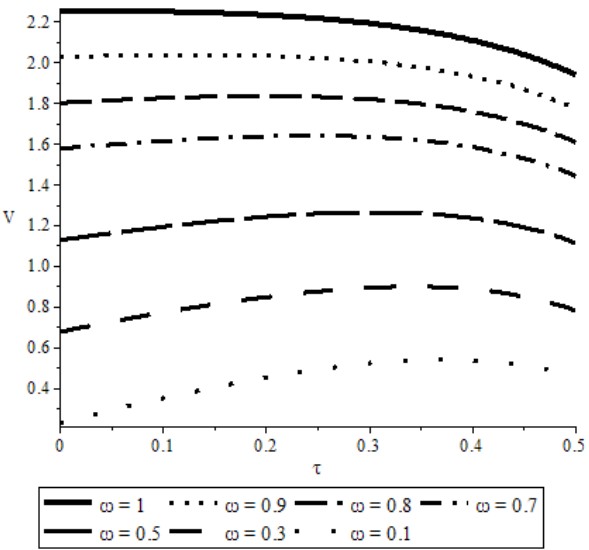

**Figure 1.** Individual utility and tax rate.

**Proposition 5.** *In a majority-voting system, the more unequal the distribution of inherited human capital, the higher will be the growth rate of the economy.*

It is obvious that the median voter prefers a higher tax rate, the greater the distance between the average wage income and her income. This means the higher the degree of the inequality of inherited human capital and accordingly of wage incomes, the higher is the tax rate in a democracy. Further, we know from Proposition 1 that a higher tax rate is associated with a higher growth rate. Hence, the more unequal the distribution of wage incomes, the higher will be the growth rate of human capital and thus of the economy, if majority voting is the governing rule in the considered economy. However, it should be mentioned that the tax rate preferred by the median voter is always lower than the growth-maximizing tax rate, because the median voter considers the income retarding effect caused by the reduction of the capital intensity.

## 7. Conclusions

In this paper we applied a simple OLG model, extended by human capital accumulation and an unequal distribution of innate human capital stock, which is inherited from parents. Additionally, we assumed that individuals can improve their human capital stock intended to increase their wage incomes as senior workers. Further, we assumed that the economy functions within a political system which follows the majority-voting rule. In this setting, we derived some interesting results. As supposed by the OECD, World Bank, European Commission and many governments, adult learning is an opportunity to increase the labor productivity and the economy's growth rate. Secondly, we have shown that a progressive tax system in our model is a way to incentivize all workers to invest in education and to reduce income inequality, although education will not reduce the inequality of gross incomes. The underlying mechanism works because the majority of voters are voting in favor of redistribution and net income increase. This is only possible in our framework by increasing the marginal tax rate, which is associated with increasing the incentive to invest in education. The motivation to invest more in education is driven by the desire to lower the individual tax payment. In the long run, all generations will benefit from this tax mechanism because of a higher growth rate of output per capita.

In general, it is also be possible to realize a Pareto-improvement because of intertemporal positive externalities associated with human capital accumulation as supposed by Lucas (1988). However, this would additionally require an intertemporal redistribution of income from the young to the old

as proposed by Boldrin and Montes (2005) and recently by Stauvermann and Kumar (2017). Further, we have shown that the more unequal the distribution of human capital, the higher will be the degree of redistribution of income and the growth rate of the economy. A main condition yielding this result is the assumption that investments in education can be deducted from the taxable income and that the individuals work in both periods of life, so that there is a clear trade-off between investments in human and physical capital.

**Acknowledgments:** Peter J. Stauvermann acknowledges thankfully the financial support of the Changwon National University 2017–2018.

**Author Contributions:** Peter J. Stauvermann and Ronald R. Kumar have both contributed 50% to the current research, and both are fully responsible for all sections because of the permanent exchange of ideas.

**Conflicts of Interest:** The authors declare no conflict of interest.

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
