# Peer review of "Adult Learning, Economic Growth and the Distribution of Income"

_economies, doi:10.3390/economies6010011_

Round 1

Reviewer 1 Report

Peer-review of the article

Adult learning, economic growth and distribution of income

The article stresses a debate concerning adult learning as an economic investment, following the human capital theory. Therefore, this article reflects on a relevant topic although not an innovative one, as this debate was first developed in the 1960’s and has benefited from several contributions since then.

In this article the authors develop a model within human capital theory stressing accumulation and inequality to derive a mechanism which may reduce the income inequality and 
provide an incentive for all adults to invest more in education. With this model the authors try to prove that the growth rate of 
the per capita income will increase and the income inequality will be reduce. 

I am not acquainted with quantitative approaches within the human capital theory. Although possessing general knowledge on the main aims of this theory, I do not feel confortable enough to debate upon the relevance of the quantitative propositions included in the model proposed by the authors. Owing to this, I will not consider the quantitative dimension of the model proposed. My review will stress the purposes of the human capital theory, which are important for understanding the relevance of this theory in present times.

The human capital theory was presented in an economic and social context of economic growth, within a Fordist accumulation pattern. The welfare state was consolidating itself in many Western countries (the so-called developed countries) and it had a strong interference in economy, fostering economic growth and distribution of income. Lifelong education (education permanente) was a relevant theoretical contribution to this frame. Analytic models such as the one presented are of relevance in this particular frame.

In spite of the importance of the Fordism accumulation pattern, present times are characterized by significant challenges: instable economic growth or even economic decrease (within recurrent and conjunctural periods of economic crisis affecting more or less countries depending on the region this country is located, the influence of international corporations, the influence - or lack of it - of public policies, etc.). Neoliberal policies have weakened the influence of the state in economy, namely in education and social welfare. The traditional pattern of economic growth and distribution of income is different than the one dominant 50 years ago. Differences in income distribution and access to lifelong learning opportunities are more expressive today than they were in the past. Following Mathews effect purposes, the more educated/qualified are more alike to join education and training, the younger ones – who in general had followed longer formal education paths – are more eager to join lifelong learning, etc. These purposes are reflected in the analytical model proposed that stresses macro trends. Lifelong learning (and not education) is to be considered as an individual tool to face professional and working challenges.

In this frame, meso analysis and micro economic analysis have to be considered. Here are some possibilities of analysis. At first stance, employment conditions of adult learners are to be considered: larger enterprises tend to develop training projects and activities and these tend to be opportunities for highly qualified workers but these can also be opportunities for less qualified ones, depending on the level of organizational change and competitive environment enterprises have to cope with. Additionally, strongly motivate workers are more alike to join lifelong learning projects and activities, being these more qualified or not. Learning has a subjective dimension that has to be enlightened. Also, in instable economic contexts like the ones many Western countries are facing (what has been called a post-crisis period), lifelong learning has not been favored by all enterprises, namely these that are medium and small. Lifelong learning is more alike to be fostered by enterprises when the economic climate is not actually bad and instable, but when it is presenting some possibilities of improving productivity and working conditions. This is a relevant fact in countries (even Western countries) that have a significant number of medium and small enterprises facing today strong international competitiveness and a lower impact of the state in economy. A last consideration refers to the shorter investments many states have been making in lifelong learning policies. As one of the outcomes, participation rates in lifelong learning in many European Union countries has not improved in the last decade. This means less lifelong learning opportunities for adult learners to be involved in lifelong learning and a higher level of motivation to join (and eventually to pay for lifelong learning activities) has to be considered.

Therefore, I believe the arguments that can be found in the article are relevant. However, due to present economic conditions and research developed in the last decades, in the introduction and conclusion authors have to refer to the importance of the readers to consider new conditions (namely the ones referred before or many others) when analyzing adult learning, economic growth and distribution of income. The model is interesting, but can not be considered just by itself as recent economic and policy developments involving lifelong learning have to be considered. This has to be mentioned in the introduction and conclusion.

Author Response

Dear reviewer,

we have added a pdf file with the answers to your comments.

Best regards  

Reviewer 2 Report

The paper can be published under certain conditions. Introduction can be shorter and more focus on the main question of the paper. Literature review could be separate from introduction to make contribution more eloquent. The wide spectrum and the number of assumptions reduce the validity of the argument. For example, it is not necessary that "the goods and factor markets are perfectly competitive" , neither that "the income of the elderly is a tax free income". Additionally, why it is necessary that  "increasing tax rate raises the investments in education on the one hand.."? Eventually, the authors should take this dimension into consideration and probably set specific research constraints at the introduction. The development of a framework of analysis could significantly contribute towards this direction. Conclusions could be better in responding to the main question as set forward at the introduction.   

Author Response

Dear reviewer,

we have prpared a pdf file with the answers we provided for the comments you made.

Best regards

Round 2

Reviewer 1 Report

The changes made by the author have made the article clearer and understandable to a larger audience. The article dully fills requirements to be published.